

# Vegetation dynamics and responses to climate anomalies in East Africa

John Musau[1,2], Sopan Patil[1], Justin Sheffield[3], Michael Marshall[2]

[1] School of Environment, Natural Resources and Geography; Bangor University, UK
[2] Climate Change Research Unit SD6, World Agroforestry Centre (ICRAF), Nairobi, Kenya
[3] Department of Civil and Environmental Engineering, Princeton University, Princeton, New Jersey

*Correspondence to*: John Musau (johnkuyega@gmail.com)

**Abstract.** Vegetation plays a key role in the global climate system via modification of the water and energy balance. Its coupling to climate is therefore important, particularly in the tropics where severe climate change impacts are expected. Consequently, understanding vegetation dynamics and response to present and projected climatic conditions for various land cover types in East Africa is vital. This study provides an assessment of the vegetation trends in East Africa using Leaf Area Index (LAI) time series for the period 1982 to 2011, regression analysis between LAI and Standardised Precipitation Evapotranspiration Index (SPEI), as well as analysis of the temporal non-stationarity in the LAI trends and vegetation response to climate. Our results show mean LAI over the region increased at a rate of about $4 \times 10^{-3}$ units per year, while the rate of increase for annual mean temperature was 0.035°C per year. Annual precipitation did not show significant trends. Trend breaks and variations in the stability of LAI time series anomalies significantly alter the LAI trends across the period of study. Drought and wetness conditions also show significant influence on the vegetation dynamics in the region. Given the potential impacts of climate change on vegetation productivity in this region, this study provides the much-needed reference point for the disentanglement of historical climatic- and human-induced vegetation dynamics. In addition, the results indicate key areas of interest for the assessment of potential impacts of vegetation dynamics on land surface water and energy balance in the region.

## 1 Introduction

Terrestrial vegetation depends on and affects land surface-atmosphere interactions as the primary link for moisture (evapotranspiration) and energy (latent) exchange through its physiological properties (Leaf Area Index (LAI), rooting depth and stomatal resistance), and its influence on surface roughness, and albedo (Arora, 2002; Bao et al., 2014; Ning et al., 2015). For instance, recent studies have reported a strong land-atmosphere coupling in West Africa, whereby vegetation dynamics play a significant role in regulating the West African monsoon and therefore rainfall distribution (Hales et al., 2006; Xue et al., 2012; Zheng and Eltahir, 1998). In South Africa, Williams and Kniveton (2012) reported increases and decreases in annual rainfall, based on idealized scenarios of expanding savanna and desert cover, respectively. Recent studies on the climatic impacts of tropical deforestation have consistently shown increased warming and reduced evapotranspiration and precipitation





(Snyder, 2010; Snyder et al., 2004). An improved characterization of spatial and temporal vegetation patterns is therefore important to not only assess landscape conditions but also to improve land surface model predictions and identify significant regional and global scale climate teleconnections.

The availability of long-term, repetitive satellite-derived datasets has greatly improved the monitoring and characterization of

the land surface at varying spatial and temporal scales. Multispectral band combinations of these datasets have aided the retrieval of long time series of land surface variables widely used to examine trends in vegetation dynamics at global, regional and national scales (Bao et al., 2014; Julien and Sobrino, 2009), impacts of vegetation on water and energy flux (Hu et al., 2009), as well as the correlation between vegetation and climate conditions (Bao et al., 2014). Particularly, LAI, which is defined in broadleaf canopies as the one-sided green leaf area per unit vegetated ground area, and in coniferous canopies as

one-half the total needle surface area per unit vegetated ground area, characterizes the physiologically functioning surface area for energy, mass and momentum exchange between the vegetated land surface and the planetary boundary layer. Hence, it is widely used by the global change research community to assess and quantify vegetation dynamics and their effects (Bobée et al., 2012; Cook and Pau, 2013; Pfeifer et al., 2014). This dataset is also a pertinent input or state variable in land surface process-based models for simulating land-atmosphere dynamics. For instance, Hu et al. (2009) used satellite-derived LAI data

to scale up estimates of evapotranspiration based on an energy balance model. Verhoef et al. (2012) used LAI data to account for the effect of canopy in calculation of surface soil heat flux. In a study by Ford and Quiring (2013), LAI was used to estimate the effects of dynamic inter-annual vegetation conditions on soil moisture, and they recommended that dynamic, rather than static, LAI parameters should be used to provide better estimates of intensity and duration of drought conditions.

Global and regional studies on the causes of variations in vegetation have shown that climatic factors, particularly precipitation

and temperature, significantly influence vegetation dynamics (Jiapaer et al., 2015; Liu et al., 2015; Montaldo et al., 2008; Tagesson et al., 2015). For instance, Tagesson et al. (2015) reported a strong link between inter-annual variation in species composition and rainfall distribution in a semiarid savanna grassland study site in West Africa region. The East Africa region, covered in this study, exhibits a wide range of climatic and ecological zones leading to diverse land cover types and land cover change dynamics (Brink et al., 2014). Due to the high dependence of livelihoods on rain-fed agriculture, there is high

vulnerability to extreme negative effects of climate change in the region (Ayana et al., 2016; Grace et al., 2014; Pricope et al., 2013). Land cover/use change is a major threat to the ecological systems in East Africa (Brink et al., 2014; Jacobson et al., 2015; Maitima et al., 2009; Pricope et al., 2013). As reported by Jacobson et al. (2015), approximately 30% of the region has been converted to cropland or urban areas with Burundi and Rwanda showing the highest proportions, 85.99% and 82.27% respectively. Between 1990 and 2010, Brink et al. (2014) found that agricultural area in East Africa (comprising Djibouti,

Eritrea, Ethiopia, Kenya, Somalia, Sudan and Uganda) increased by 28% with an alarming shift in the rate of deforestation from 0.2% per year in 1990-2000 period to 0.4% per year in 2000-2010 period. Pricope et al. (2013) addressed the spatial interaction between climate, vegetation variations and degradation, and population density changes in the East Africa Horn's pastoral and agro-pastoral livelihoods zones. They established a potential long-term degradation of rangelands mainly due to population pressures and land use change.



One of the most disastrous and damaging hazards in East Africa is drought. As noted by Ayana et al. (2016), the frequency of drought in the greater horn of Africa has doubled from once every 6 years to once every 3 years, and has partly contributed to the increase in resource-based conflicts in the region. Despite the central role of combined precipitation and temperature effects on vegetation productivity in East Africa, the vegetation trends and the vegetation-climatic relationships across the East Africa

region have not been adequately evaluated based on the readily available LAI data set covering the period 1982-2011. Investigation of the vegetation trends and its response to the precipitation and temperature conditions across the region will help in better understanding of the impacts on terrestrial ecosystems and identification of emerging vulnerable areas in the region. This is vital for better planning and management to mitigate ecological and economic loss. This study seeks to shed new light on vegetation trends and responses to climate anomalies across East Africa and in different land cover types in the

region. In addition, we have evaluated the impacts of biogeographical factors on vegetation response to combined precipitation and temperature index. Specifically, this study aims to: (i) investigate spatiotemporal patterns of long-term vegetation trends based on LAI dataset for the period 1982-2011 at 8 km spatial resolution; (ii) evaluate vegetation response to a simple multi-scalar drought index (the Standardised Precipitation-Evaporation Index (SPEI, Vicente-Serrano et al., 2010) that combines precipitation and temperature data at varying time scales, and (iii) understand the relations between vegetation responses to

SPEI and biogeographical factors.

## 2 Study area and data

### 2.1 Study area

Our study area spans 2,267,136 km$^2$ (bounded by 5.52ºN and 11.76ºS latitude, 28.8ºW and 41.92ºE longitude) and comprises the countries of Burundi, Kenya, Tanzania, Rwanda and Uganda, and portions of the Congo, Ethiopia, Malawi, Mozambique,

Somalia, South Sudan and Zambia (Figure 1). A broad overview of the relation between climate and the key vegetation zones in the region is described in White (1983). The northward migration of the Intertropical Convergence Zone (ITCZ) in the summer months initiates a bimodal precipitation pattern in the majority of the region with a main rain season during March to May and short (monsoonal) rains during October to November (McNally et al., 2016). The Somalia-Masai ecoregion covers most of Kenya between the highlands and coastal belt as well as the dry lowlands of north and central Tanzania. This ecoregion

consists mainly of arid and semi-arid climate with a mean annual rainfall less than 500 mm and high mean monthly temperature of between 25°C and 30°C.

The Sudanian ecoregion extends from South Sudan to West Uganda and it is mainly characterized by a semi-arid and equatorial savanna type of climate with a severe dry season. The highlands and mountain areas of Kenya as well as most of the southern and western parts of Uganda, with more than 1000 mm mean annual rainfall in the forest zone are defined as Afromontane.

Along the Kenyan, Tanzania and Southern Somalia coastline is the Zanzibar-Inhambane ecoregion, which consists of forests and Mangroves and is characterised by mean annual rainfall between 800 and 1200 mm. Most parts of Uganda, and some parts



of western Kenya, northern Tanzania and Eastern Congo as well as the whole of Eastern Rwanda and Burundi comprise the Lake Victoria ecoregion, which is characterized by rain forest with semi-evergreen forest and woodland/shrubland as the dominant vegetation type.

## 2.2 Data

We used the third generation Global Inventory Monitoring and Modelling Studies LAI (GIMMS LAI3g) dataset spanning the period 1982 to 2011, at approximately 8 km spatial resolution and a 15-day interval, to characterize vegetation dynamics. As described in Zhu et al. (2013), the dataset was produced by the fusion of GIMMS NDVI3g (Pinzon and Tucker, 2014) and an improved version of the Moderate Resolution Imaging Spectroradiometer (MODIS) LAI using a Feed Forward Neural Network (FFNN) algorithm. The GIMMS LAI3g data has been validated using ground based observations measured across

East Africa (Pfeifer et al., 2014) and has also been used to study vegetation dynamics at a global scale (Cook and Pau, 2013). To remove superfluous values in the data, the biweekly LAI data were smoothed using an optimized Savitzky-Golay (S-G) filter commonly used to correct Earth observation data (Chen et al., 2004). The smoothed biweekly dataset was then aggregated using the maximum value composites (MVC) approach to create a monthly LAI time series from 1982 to 2011. The MVC approach obtains monthly values as the maximum value per pixel in each pair of bi-monthly datasets.

The climatic data sets used included precipitation, minimum and maximum temperature. The precipitation data was obtained from version 2 of the Climate Hazards group Infrared Precipitation with Stations (CHIRPS) dataset (Funk et al., 2015). The CHIRPS dataset is a 0.05° (~5 km) spatial resolution global gridded dataset of daily precipitation available from 1981 to 2015. It is obtained by merging satellite observations, weighted average precipitation from stations for a given pixel, and precipitation predictors such as elevation, latitude and longitude (Funk et al., 2015). It has been compared with other satellite precipitation

estimates and observed rain gauge data (Ceccherini et al., 2015; Dembélé and Zwart, 2016; Toté et al., 2015) and has also been used in previous studies in East Africa (Ayana et al., 2016; Pricope et al., 2013). Minimum and maximum air temperature data were obtained from a high resolution daily meteorological dataset developed by the Princeton university hydrology group for East Africa (Chaney et al., 2014; Sheffield et al., 2006). The datasets were also resampled to 8km spatial resolution at a monthly time-step.

The Synergetic land cover product (SYNMAP) (Jung et al., 2006) was used in this study to delineate major land cover classes. This is an improved global land cover product reflecting global land covers around year 2000 at 1-km spatial resolution and consisting of 48 different classes. This dataset was selected particularly for this study as it covers a period approximately at the middle of our study period. The dataset is derived based on fuzzy agreement of different global land cover products, including, Global Land Cover Characterization Database (GLCC), GLC2000, and the 2001 MODIS land cover product, with

consideration of individual strengths and weaknesses of mapping approaches. The main land cover types in our study area include evergreen broadleaf forest (EBF), deciduous broadleaf forest (DBF), mixed forest (MF), shrubs, grasses, crops and bare areas (Figure 1). In this study, dynamic land cover changes were not considered, which may bring in uncertainties due to change of vegetation type and land use activities. The land cover data was reclassified in to six main classes namely forest



(evergreen broadleaf trees, deciduous broadleaf trees and mixed trees), shrubland (shrubs, trees-shrubs mosaic, and shrub-barren mosaic), wooded grassland (trees-grasses mosaic, and shrubs-grasses mosaic), grassland (grasses), cropland (cropland and cropland/natural vegetation mosaic) and bare areas. The reclassified land cover data was then aggregated to grid cells of approximately 8km x 8km to be consistent with the resolution of LAI and climate datasets. The Global Human Footprint Index dataset (LWP-2) was used as a proxy for anthropogenic effects (WCS and CIESIN, 2005) to assess the human influence on vegetation dynamics and response to climatic conditions.

## 3 Methods

### 3.1 Trend analysis

#### 3.1.1 Long-term trend analysis

We applied the Mann-Kendall (MK) trend test method to verify the existence and direction of significant long-term trends in the data, and Thiel-Sen median slope estimator (TSE) to quantify the strength of the trend. The MK test is a non-parametric method which measures the degree to which a trend is a monotonic increase or decrease over time. Kendall's τ ranges from -1 to 1 where -1 indicates a consistently decreasing trend while 1 indicates a consistently increasing trend and zero indicates no consistent trend. The MK test for the statistical significance ($p<0.05$) of Kendall's τ was considered appropriate since the assumption of normality in data distribution does not affect its validity. The TSE, used to quantify the strength of a trend, computes the trend as the median of the slopes between all $n(n − 1)/2$ pair wise combinations over time. It is a rank-based regression approach and is resistant to outliers. Its wide application has demonstrated good potential in estimating trends in vegetation and climatic time series data (Fensholt et al., 2012, 2013; Marshall et al., 2012; Teferi et al., 2015).

The serial correlation in high frequency time series data (daily, weekly or monthly) has been shown as a major challenge in long-term trend analysis due to its effects on trend overestimation and detection even when there is none, thereby creating false rejections of the null hypothesis of a trend test (Wang et al., 2015; Yue and Wang, 2002). To address this, we opted to avoid seasonality in the time series data by using yearly aggregated data, which has been suggested inprevious studies (Boschetti et al., 2013; de Jong et al., 2011; de Jong and de Bruin, 2012). Furthermore, the Trend Free Pre-whitening (TFPW) procedure proposed by Yue et al. (2002) was used remove serial correlation from the time series based on a lag-one autoregressive model. In this procedure, if the slope estimated by the TSE is not equal to zero, a linear trend is removed from the data. A lag-1 serial correlation coefficient of the de-trended data is then computed and the AR(1) is removed. The pre-whitened residuals and the initially estimated trend are then blended, and the MK test is applied to the blended series to measure the significance of the trend.

We used monthly LAI values as a proxy for vegetation dynamics in the region. This dataset has been used previously to investigate long-term vegetation trends (Cook and Pau, 2013). The long-term trends were analysed on the region-wide averaged data and per-pixel on annual and seasonal basis. The two main rain seasons in the region (long rains: March to May



(MAM) and short rains: October to December (OND)) were considered. To obtain the area-averaged data, first we calculated per-pixel annual and seasonal mean temperature, total precipitation and maximum LAI values. Anomalies of LAI, precipitation and temperature were then calculated against 30-year mean. The resultant per-pixel anomalies for each dataset were then averaged to obtain region-wide average time series which were used for long-term trend analysis. Although region-wide trends

provide useful information about changes in vegetation and climatic condition, they do not reflect the spatial inconsistencies within the region. Consequently, we also evaluated the spatial heterogeneity in the long-term vegetation trends based on per-pixel analysis of LAI anomalies.

### 3.1.2 Temporal non-stationarity of LAI trends

We used the Breaks For Additive Season and Trend (BFAST) algorithm (Verbesselt et al., 2010) to identify shifts in the trend

and seasonal components of the LAI time series. This algorithm iteratively splits the time series into seasonal, trend, and residual components, while trend and seasonal breakpoints and their associated confidence intervals are estimated for the seasonality and trend components. This allows extraction of the anomaly time series while explicitly accounting for the non-stationarity (gradual and abrupt changes) in the trend and seasonal components of the time series. Based on the information output by BFAST, the largest magnitude break was detected, and its sign was used to characterize the non-stationarity of LAI

trends. These trends were categorized into the following 6 classes: (i) monotonic increase, (ii) monotonic decrease, (iii) greening with a setback (increase with negative break), (vi) browning with a burst (decrease with positive break), (vii) reversal: increase to decrease, (viii) reversal: decrease to increase (De Jong et al., 2013).

### 3.2 Vegetation response to climatic conditions

### 3.2.1 Characterising drought/wetness conditions

We used the Standardized Precipitation-Evaporation Index (SPEI), which is based on precipitation and Potential Evapotranspiration (PET) data, to characterise the drought/wetness conditions in a given area at dynamic time-scales. Compared to climatic indices based on precipitation or temperature data alone, SPEI is considered a superior climatic indicator as it considers the effect of temperature on water balance through its influence on the atmospheric evaporative demand. SPEI is multi-scalar, and can therefore be calculated at a range of time-scales (1 to 48 months) to assess water deficit impacts at

short- and long-time scales. A user-defined calibration period (reference period) is used to calculate the average water balance while the deviations from this average are determined at varying time-scales. Positive SPEI values represent wet conditions, whereas negative values represent drought conditions. Due to differences in physiological or edaphic factors, some vegetation types may respond to short-term soil water deficit periods, while others may be more resistant and only respond to soil water deficits of longer durations. Therefore, at a regional scale, time scales of optimum SPEI-vegetation correlation are expected to

vary spatially (Vicente-Serrano et al., 2013). We used the climate data described in section 2.2 to compute PET based on a modified-Hargreaves (MH) method, which includes a rainfall term (Droogers and Allen, 2002; Hargreaves, 1994). SPEI was




then estimated using the climatic water balance defined as precipitation minus PET  (Vicente-Serrano et al., 2010, 2013) in R software using the SPEI package.

### 3.2.2 Short-term vegetation response to climate

We analysed vegetation response to climatic conditions using LAI anomaly obtained from BFAST analysis to account for the breakpoints in the trend and seasonal components of the time series.  In addition, SPEI obtained at a three-month timescale (i.e. SPEI calculated on cumulative water balance over previous 3 months) was used. Although the maximum LAI-SPEI correlation is characterised by variations in the SPEI timescales in different vegetation types, we used a three-month time-scale to assess the short-term vegetation response. Following De Keersmaecker et al., (2015 and 2017), three response metrics were used to described the short-term vegetation response: (i) variance metric (the standard deviation of the LAI anomaly time series); (ii) resistance metric (the association between the LAI anomaly and SPEI time series); and (iii) resilience metric (the auto-correlation at lag one of the LAI anomaly).

To obtain the latter two metrics, we used a linear relationship between monthly LAI anomalies and SPEI at three-month timescale defined as follows.

$$LAI_t = \alpha \cdot SPEI_t + \phi \cdot LAI_{t-1} + \varepsilon_t$$

Where $LAI_t$ is the standardised LAI anomaly at time $t$, $SPEI_t$ the standardized SPEI at time $t$, and $\varepsilon_t$ is the residual term at time $t$. $\alpha$ and $\phi$ are the model's coefficients. α is an indicator of the extent to which the vegetation deviates from its equilibrium due to droughts anomalies, thus expressing the resistance against drought. Similarly, $\phi$ relates to vegetation resilience as it gives an indication of the dependency of the anomalies on previous values. Large absolute values of $\alpha$ indicate a low resistance to droughts anomalies, hence a large vegetation response to short term droughts anomalies. On the other hand, large absolute values of ϕ imply that the anomalies are strongly determined by the anomaly at time $t-1$ and indicate a low resilience, i.e. a slow return to ecosystem equilibrium after disturbance. The time series were standardized to assure comparability between the model coefficients.

In addition to the response metrics obtained for the entire study period, a twelve-year moving window was used to obtain time series of response metrics. The trend of these time series (obtained using the non-parametric Kendall τ rank correlation coefficient) was used to define the temporal non-Stationarity of the short-term vegetation response to climatic conditions. To reveal the climatic impacts on LAI variance metric, we applied a similar approach on monthly SPEI data to obtain the time series of climatic variance. We further calculated the Kendal τ rank correlation coefficient between the vegetation and climatic variance time series.

As noted by Hawinkel et al. (2016), vegetation response to climate variability in East Africa is influenced by a set of biogeographical factors. We therefore analysed the spatial variations in the vegetation response metrics based on their relationship with the annual average water balance and Human Footprint Index. As the vegetation response is not linearly related to all explanatory factors, we used a generalized additive model with integrated smoothness estimation (Hastie and





Tibshirani, 1990). The global effect of these factors on vegetation response is modelled using data across the region while local effects are analysed per land cover type.

# 4 Results

## 4.1 Trend analysis

### 4.1.1 Long-term trends in LAI and climatic conditions

When averaged across the region, both mean LAI and temperature showed significant increasing trends during the period 1982-2011, while total precipitation showed no significant trend during the study period (Table 1). Mean LAI increased at a rate of about $4\times10^{-3}$ per year while the rate of increase for annual mean temperature was 0.035°C per year. On a seasonal basis, average temperature increased significantly in the OND season at a rate of 0.036°C per year, while LAI and precipitation did not show significant trends. In the MAM season, LAI and average temperature increased significantly while precipitation did not show significant trend.

Figure 2 shows the spatial heterogeneity in LAI trends. Considering only the statistically significant pixels ($p \leq 0.05$) and the total vegetated area in the region, the increasing and decreasing annual LAI trends accounted for 25.37% and 3.94% respectively. During the MAM season, positive trends showed a wider coverage at 31.04% compared to 3.87% for the negative trends. Compared to annual and MAM trends, the OND season shows more widespread declining vegetation trends at 12.68% while positive trends covered 18.91% of the area.

Northern parts of Kenya show significant negative LAI trends, while increasing trends are prevalent in the East Sudanese Savanna (extending from South Sudan to North Uganda and mainly composed of trees and shrub cover) and the southern parts of Tanzania (mainly covered by deciduous broadleaf and mixed trees) for annual and MAM season. During the OND season, negative trends are prevalent in the deciduous broadleaf and mixed tree covered areas in Tanzania and Malawi. Along the coast region of Kenya (mainly composed of Evergreen broadleaf trees, tree/grass mosaic and cropland) and the East Sudanese Savanna significant positive trends were prevalent during the OND season.

## 4.1.2 Temporal non-stationarity of LAI trends

Based on the BFAST trend analysis, about 78.3% of the study area showed statistically significant (p<0.05) LAI changes for the study period (Figure 3). As shown in Figure 3, about 73.2% of the entire study area (or approx. 93.5% of all cases of significant LAI changes) indicated abrupt changes (composed of the interrupted and reversed trend classes) in the LAI timeseries. Pixels with interrupted trends accounted for 46.9% (composed of 26.6% showing greening with a setback while 20.3% showed browning with a burst) of the area with vegetation cover. In comparison, reversed trends were identified in 26.3% of the region, composed of greening to browning in 18.4% and browning to greening in 7.8%. On the other hand, 5.19% of the study area showed monotonic greening (4.72%) and monotonic browning (0.47%). The observed trend types in the





region were therefore dominated by, in a descending order, increasing trend with negative break, decreasing trend with positive break, reversed increase (increase to decrease) and reversed decrease (decrease to increase).

Large patches of decreasing trend with a positive break were particularly noted in the North-eastern Kenya and Tanzania, areas mainly covered by grass and xeric shrubs. Interruptions of decreasing trends were mainly recorded in the 1993-1997 period.

Majority of the areas with significant change in both segments was characterized by an increasing trend with a negative break. Large areas showing significant change only in the second segment mainly showed a decreasing trend with a positive break while their timing of the break was predominantly 1993 – 1997 for Kenya and after 2002 for Tanzania. When compared across the region, these two classes of timing of trend shifts appeared to be the most common. A detailed analysis of significance of the trend segments showed that more than 25% of the respective total coverage of cropland, forest, wooded grassland and

shrubland showed significant trends in both segments or no break and significant change. In addition, irrespective of the land cover type, pixels with significant change in only one of the two segments often showed significant trend in the second segment. This analysis also revealed that major changes observed in the LAI trends across the region occurred recently. Irrespective of the land cover type, more than 35% of the shifts in the LAI dynamics were noted in the period after 2002 while the periods before 1988 and between 1988 and 1992 are characterized by the lowest proportions of the detected trend shifts at

3.02% and 10.7%, respectively. Trends shifts in the 1998 – 2002 period were predominantly composed of increasing trend with negative break and reversed increasing trend (increase to decrease).

Across the different land cover types, high proportion of pixels indicated increasing trend with a negative break (5), decreasing trend with a positive break (6), and reversed increasing trend (increase to decrease) (7). Shrubland, which constitute the majority land cover type in the region, showed widespread monotonic greening (break types 1 and 3) compared to other land

cover types. A greater proportion of interrupted trends and a comparable proportion of pixels with reversed trends were found in cropland. Reversed greening and browning is predominant in grassland.

## 4.3 Vegetation response to climatic conditions

### 4.3.1 Regional average climatic water balance

Figure 4 shows the spatial pattern of long-term average water balance (i.e precipitation minus PET) across East Africa. The

25 water balance shows values increasing from the north-east to the south-west of the region. In northwest Kenya the long-term average water balance is below -1500 mm being a typical arid area also characterised by rock outcrops and bare areas. Semi-arid areas are shown extending from south Ethiopia through central Kenya into north and central Tanzania. These areas are mainly composed of grasslands, cropland, wooded grassland and shrubland. The humid and semi-humid areas are found in west of the region, western Kenya and southern parts of the region. These areas are mainly composed of forests,

tress/shrub/grass mosaics and cropland. We selected twelve regions representatively, shown by solid boxes in Figure 4, for case studies on LAI-SPEI correlation at various timescales.



### 4.3.2 Short-term vegetation response to climatic conditions

Although three-month time scale was selected for vegetation response analysis in our study, the maximum LAI-SPEI correlation is expected to occur at varying timescales across the region. Figure 5 shows the variations in LAI-SPEI correlation at different time-scales for the selected case study areas (shown in Figure 4). While these selected locations represent different water balance regions in the study area, they also coincide with different land cover types. As shown in this figure, differing vegetation response to SPEI time-scales is evident in different water balance regions. Particularly, locations b, d, e, f, g and i show stronger positive LAI-SPEI correlation while locations a, k and l shows strong negative LAI-SPEI correlation. Weak correlation is shown in locations c, h, and j at varying time-scales. Location a, which comprises of trees-grass mosaic is characterised by prevalent negative correlations while potential weak positive relationship is indicated during the March-May season across all the time-scales. Both locations c and j, which are characterised by semi-arid and semi-humid climatic conditions as well as shrubs and trees-shrub mosaic land covert types, respectively, do not show distinctive patterns in the LAI-SPEI correlations. Locations f and h, which are respectively covered by crop-vegetation mosaic and grassland, showed a similar pattern in the LAI-SPEI correlation characterised by low correlation values in the May-August period.

Figure 6(a) shows the increase/decrease in LAI anomaly variance given by the Kendall τ coefficient for the standard deviation derived over a twelve-year running window. The trend in vegetation variance/stability is positive and statistically significant in most parts of the region. Figure 6(b) shows the Kendall τ rank correlation coefficient between the LAI and SPEI variance time series. As shown in this figure, variations in vegetation stability can be linked to climatic conditions. Most of the pixels indicated a positive relationship between the LAI and SPEI variances which implies that an increase/decrease in vegetation variance is linked to increase/decrease in climate variability. Stronger positive trend in vegetation variance shows a similar spatial pattern compared to the LAI-SPEI stability correlation, implying widespread influence of precipitation on vegetation trends in the region.

A correlation analysis between the two metrics (i.e. Kendall τ coefficient for LAI variance time series and Kendall τ rank correlation coefficient between the LAI and SPEI variance time series) showed that, although not strong (r = 0.44), the spatial relationship between both coefficients was positive and significant. This indicates that positive SPEI variance trends tend to favour positive LAI variance trends across the region. The spatial variations in vegetation stability and relationship between vegetation and SPEI variance also reflect differences in land cover types. The LAI variance shows widespread increasing trend across all the land cover types but predominantly in the grasslands and wooded grasslands. A similar pattern is indicated in the correlation between LAI variance and SPEI variance. However, a decrease in this correlation is prevalent in forests, shrubland and cropland which also showed large proportions of decreasing trend in LAI variance.

Figure 7 shows the spatial distribution of vegetation drought-resistance and resilience coefficients over the complete period (1982 to 2011). Although the model converged effectively with RMSE <0.9 in all pixels, coefficients were not significant in some pixels at 95% confidence level, which were masked from the analysis. Vegetation drought-resistance coefficients were positive and largely significant, emphasizing the predominance of water balance forcing on vegetation in the region. The



spatial distribution of this coefficient generally reflects the spatial patterns of the different land cover types in the region. High and significant drought-resistance coefficient sis evident in the stretch extending from south-eastern area of South Sudan to east of Uganda and western Kenya into Norther parts of Tanzania. This is indicative of the low resistance thus large vegetation response to short term drought anomalies in these areas. These areas are mainly composed of grassland, cropland and

crop/natural vegetation mosaic land cover types. However, some areas showed insignificant drought-resistance coefficients, mainly in western and southern parts of the region that composed of deciduous and evergreen broadleaf and mixed tree cover. On the other hand, resilience coefficients were positive and significant across the region. High vegetation resilience coefficients were prevalent in Kenya, Tanzania and eastern parts of Uganda, which implies slow return to ecosystem equilibrium after potential disturbance in those areas. In addition, the two coefficients (vegetation resistance and resilience) showed widespread

contrast in their spatial distributions. The areas with low drought-resistance coefficients (i.e. high resistance to drought) also show high resilience coefficients (i.e. low resilience) and vice versa. For instance, the north-eastern Kenya region (mainly composed of grassland and shrubland) showed low drought resistance coefficient and a high resilience coefficient.

The sensitivity of vegetation response to water balance and human footprint index in different land cover types was compared to the regional sensitivity across East Africa (Figure 8). Across the region, vegetation response to climatic conditions is most

strongly determined by the climatic conditions, human factors as well as structural features of the vegetation itself. As shown in Figure 8a and b, vegetation resistance coefficient is significantly and negatively related to the annual water balance across the region which shows that vegetation in the low water balance areas is more sensitive to drought anomalies compared to relatively humid areas. The sensitivity of the resistance coefficient in different land cover types across the region also shows significant variations. Areas dominated by herbaceous vegetation cover (wooded grassland, grassland and croplands) show

larger overall sensitivity to short-term SPEI anomalies in arid and humid areas. Particularly, cropland show higher sensitivity compared to the regional curve in the areas with annual water balance less than -750 mm and greater than -200 mm while resistance in grassland shows high sensitivity beyond -750 mm of annual water balance (Figure 8a). On the other hand, drought resistance in grassland and wooded grassland shows a higher sensitivity to human influence while cropland shows a lower sensitivity compared to the regional curve. In shrublands, the impact of annual average water balance on vegetation resistance

approaches the average regional curve with a decreasing sensitivity beyond -250 mm of annual water balance. This land cover type also shows a lower sensitivity to human influence below 35% of the human footprint index. Drought resistance in forests shows a consistently lower sensitivity to both annual water balance and human influence.

On the other hand, vegetation resilience coefficient is negatively related to both annual water balance and human footprint index in the region (Figure 8c and d). Unlike the resistance coefficient, the sensitivity of vegetation resilience to both factors

does not vary widely across different land cover types. Except in cropland, the different land cover types show a rapid decreasing sensitivity of vegetation resilience coefficient to water balance below -1000 mm. Both shrubland and wooded grassland show a relatively complicated sensitivity of resilience to water balance. On the other hand, except in cropland, human influence on vegetation resilience is relatively higher in other land cover types compared to the regional curve. Sensitivity of forest resilience is fairly constant across the region.



In addition to the vegetation response across the complete period of analysis, the temporal changes in the drought resistance and resilience coefficients were also analysed. Figure 9 shows the spatial heterogeneity in the temporal variations of vegetation drought-resistance and vegetation resilience coefficients. The vegetation resistance metric shows the largest increase in forest, wooded grassland and grasslands. These land cover types are also characterised by the largest increase in the resilience metric.

Forests and cropland showed the highest spatial variance in the trend of resistance metric while the trend of resilience metric varied widely in wooded grassland and grasslands.

## 5 Discussion

### 5.1 Spatio-temporal variations in vegetation

The east Africa region, which is mainly characterised by vast dryland ecosystems, was focused in this study. These ecosystems

are often over-utilised for pastoral grazing and mixed cereal cropping systems thus exacerbating their vulnerability to extended drought occurrences leading to severe negative implications on food security and community livelihoods (Hoscilo et al., 2015; Landmann and Dubovyk, 2013; Pricope et al., 2013). The results presented here provide a view of vegetation dynamics that could be used to fully appreciate where significant changes in ecosystem functioning have occurred in the region. Vegetation trend analysis using the GIMMS LAI showed a significant increase in the annual vegetation condition in over 25.37% of study

area for the period 1982–2011. In northern parts of Uganda and DRC, increasing LAI trends were found in the annual and MAM time series. These areas are characterised by savanna-forest transition land cover types mainly composed of grassland, shrubland and wooded grassland. These areas were also identified by Mueller et al. (2014), in a global NDVI trend analysis, as part of eco-regional extremes for NDVI increase. This increase in LAI could be attributed to increasing land cover transition to croplands. The southern parts of Tanzania, particularly the Tanzania's Eastern Arc mountain ranges, also showed increasing

LAI trends in the annual and MAM time series. Widespread decreasing LAI trend found in Tanzania during the OND season coincides with Vrieling et al. (2013)'s finding of a decrease in the length of growing season. The significant and persistent negative trends in north-central and southern Kenya coincide with a significant decline in precipitation and can be attributed to climatic effects, as also reported by Hoscilo et al. (2015). In addition, the decline in LAI shown in our analysis could also be a combined effect of climate as well as replacement of shrubs by grass and crops with lower LAI values in areas

characterised by intensive pastoral activities.

The temporal non-stationarity of LAI trends derived over the complete study period varies spatially and depends on the land cover type. Although the regional variations in LAI are closely linked to climatic and human-induced factors, it is still unclear how the increasing and decreasing trends shown from the timeseries analysis are influenced by the different land-use changes in the region. A wide coverage of significant reversed increasing trends (increase to decrease), particularly in Kenya and

Tanzania, coincides with both significant and non-significant decreasing trend in the long-term trend analysis. This indicates the need to consider potential turning points in long-term vegetation index time series analysis. Interrupted positive trends (increase with negative break) across the region leads to a decline in areas identified as indicating increase in long-term trends.



## 5.2 Vegetation response to climate

The stability of natural and productive ecosystems and their flow of services is crucial especially amid potential climate change impacts. Assessment and quantification of this stability has been largely aided by the availability of regional to global scale and long-term time series of vegetation indices derived from readily available remote sensing datasets. The vegetation response
metrics derived in this study revealed contrasting spatial patterns. For instance, a sample set of pixels representing different land cover types under varied water balance regions across the study area showed highest LAI-SPEI monthly correlations at various time-scales.

The vegetation variance, resistance, and resilience metrics obtained across the study period showed widespread spatial heterogeneity, which indicates the influence of land cover types on vegetation response to short-term droughts and memory
effects. The fact that vegetation response is stronger for a given range of annual water balance emphasises also the effects of different vegetation formations. The vegetation resistance coefficient in forest environments was evidently very small and statistically insignificant, which implies that the greenness of trees is not largely influenced by short-term variations in the water balance. This corresponds with the findings of Camberlin et al. (2007) based on NDVI-rainfall regression analysis in tropical Africa. In addition to the lack of vegetation response to inter-annual water balance variability, seasonal LAI variations
in most of these areas do not match seasonal rainfall variability. In these areas, leafing can be induced by rainfall amounts even lower than average while the effects of moisture deficit are hampered by the capability of the vegetation to tap deep soil water resources. In addition, the lack of significant vegetation response in such vegetation formations may be attributed to other biases in the LAI time series such as cloud contamination as well as predominance of other vegetation growth constraints (Huxman et al., 2004) .

The annual average water balance emerged as the key factor determining the level of vegetation resistance to drought anomalies compared to the human footprint. A high sensitivity of vegetation resistance coefficient across the region coincides with intermediate water balance areas (-1000 to -500 mm). The major peak of the vegetation resilience sensitivity to water balance is shown at -750 mm. This relates with findings of Huxman et al. (2004) based on the correlation analysis of net primary production and annual precipitation data at sites sampled from major global biomes. However, the influence of annual average
water balance on vegetation resistance coefficient is somewhat intricate: a positive effect is shown in areas with annual average water balance below -750 mm, which changes to decreasing influence between -750 mm and 0 mm and then to relatively constant sensitivity in areas with annual average water balance greater than 0 mm. The low sensitivity in drier areas has been linked to vegetation drought resistance strategies such as low specific leaf area, high root–shoot ratio and low stomatal conductance (Paruelo et al., 1999). While in wetter areas, the vegetation is also well-adapted to the temporary seasonal
constraint in water availability (Camberlin et al., 2007).

In addition to response metrics derived across the complete study period, this study also quantified the magnitude and direction of temporal vegetation response changes in east Africa. The temporal changes in the vegetation response metrics imply technical and ecological effects. Therefore, the assumption of stationarity in whole time series is not realistic for the analysis



of vegetation dynamics. In addition, results are likely to differ significantly depending on the time series length as well the data sources (De Keersmaecker et al., 2017). However, further analysis is required to disentangle human and climatic induced causes of these variations.

## 6 Conclusions

This paper focused on understanding the spatial-temporal variations in LAI during 1982-2011 period over East Africa based on robust non-parametric trend tests. We found hotspots with significant LAI declines over the last 30 years, thus signifying the areas of potential land degradation and increased vulnerability to climate change in the future. Although potential climatic degradation has been cited in these areas, other factors such as population pressures and declining land health should be considered in future studies in the region. The region is mainly characterized by sparse vegetation that is composed of grass

and shrubs. At the 8km spatial resolution used in this study, some gradual and abrupt vegetation changes may have been masked. We therefore recommend further analysis at higher spatial resolution. The BFAST decomposition is a useful approach for the detection of abrupt intra-annual changes within the trend and seasonal components and their time of occurrence, as well as the quantification of the magnitude of these abrupt changes detected during the study period. This approach provides valuable support in decision-making on potential ecosystem degradation hot-spots and further unravelling of human and

climatic related disturbances to ecosystem functioning.

The vegetation-climate regression analysis provided a view of the interactions between vegetation and climate. However, there is need for further analysis of the multifaceted connection between vegetation production patterns to human and climatic drivers in region to account for the individual and coupled effects of both natural and anthropogenic determinants of terrestrial ecosystem functioning. This can be achieved through studies incorporating long-term climate change, variations in climatic

extremes and $CO_2$ fertilization, as well as potential land-atmosphere feedbacks of land use/cover changes and increased human footprint. Future studies in this region should also attempt to explore vegetation responses based on the use of well parameterized dynamic vegetation growth models that solve the land surface water and energy balance in a coupled manner. It is also worth noting that higher resolution LAI and climatic data might present a clearer picture of the vegetation dynamics since a large proportion of the study area is mainly arid and semi-arid thus at 8 km spatial resolution, the actual vegetation

dynamics may not be well captured.

*Acknowledgment:* This work was supported primarily by the CGIAR research program on Forest, Trees, and Agroforestry under the project, titled: *Earth Observation Based Assessment of the Water Balance in East Africa*.

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





Table 1: The trends in annual and seasonal mean LAI, temperature and precipitation in the period 1982–2011 for different land cover types and across the whole region. The slope indicates change per year while significance is defined as $+ = p < .1$, $* = p < .05$, $** = p < .01$, $*** = p < .001$

| | | Forest | | Shrubland | | Wooded grassland | | Grassland | | Cropland | | Regional | |
|---|---|---|---|---|---|---|---|---|---|---|---|---|---|
| | | Slope | Tau | Slope | Tau | Slope | Tau | Slope | Tau | Slope | Tau | Slope | Tau |
| Annual | LAI | 0.005 | 0.325 * | 0.004 | 0.345 ** | 0.003 | 0.251 + | 0.002 | 0.143 | 0.005 | 0.291 * | 0.004 | 0.310 * |
| | Temp | 0.034 | 0.537 *** | 0.039 | 0.532 *** | 0.037 | 0.512 *** | 0.036 | 0.512 *** | 0.037 | 0.527 *** | 0.036 | 0.522 *** |
| | Precip | -0.225 | -0.330 * | -0.124 | -0.084 | -0.080 | -0.034 | -0.085 | 0.020 | -0.069 | -0.020 | -0.116 | -0.118 |
| MAM | LAI | 0.010 | 0.512 *** | 0.007 | 0.404 ** | 0.004 | 0.172 | 0.006 | 0.177 | 0.009 | 0.414 ** | 0.007 | 0.389 ** |
| | Temp | 0.023 | 0.330 * | 0.035 | 0.468 *** | 0.035 | 0.409 ** | 0.029 | 0.335 * | 0.029 | 0.419 ** | 0.029 | 0.399 ** |
| | Precip | -0.497 | -0.192 | -0.456 | -0.192 | -0.519 | -0.182 | -0.286 | -0.064 | -0.357 | -0.108 | -0.486 | -0.158 |
| OND | LAI | -0.001 | -0.138 | 0.002 | 0.133 | 0.005 | 0.286 * | 0.003 | 0.232 + | 0.002 | 0.128 | 0.002 | 0.158 |
| | Temp | 0.037 | 0.571 *** | 0.038 | 0.527 *** | 0.032 | 0.493 *** | 0.034 | 0.542 *** | 0.039 | 0.562 *** | 0.036 | 0.542 *** |
| | Precip | -0.616 | -0.138 | -0.131 | -0.030 | 0.135 | 0.089 | -0.199 | 0.015 | -0.293 | -0.044 | -0.286 | -0.015 |



**Figure 1: Location of the study area and land cover types based on the Synergetic land cover product (SYNMAP) at 1-km spatial**
5   **resolution**



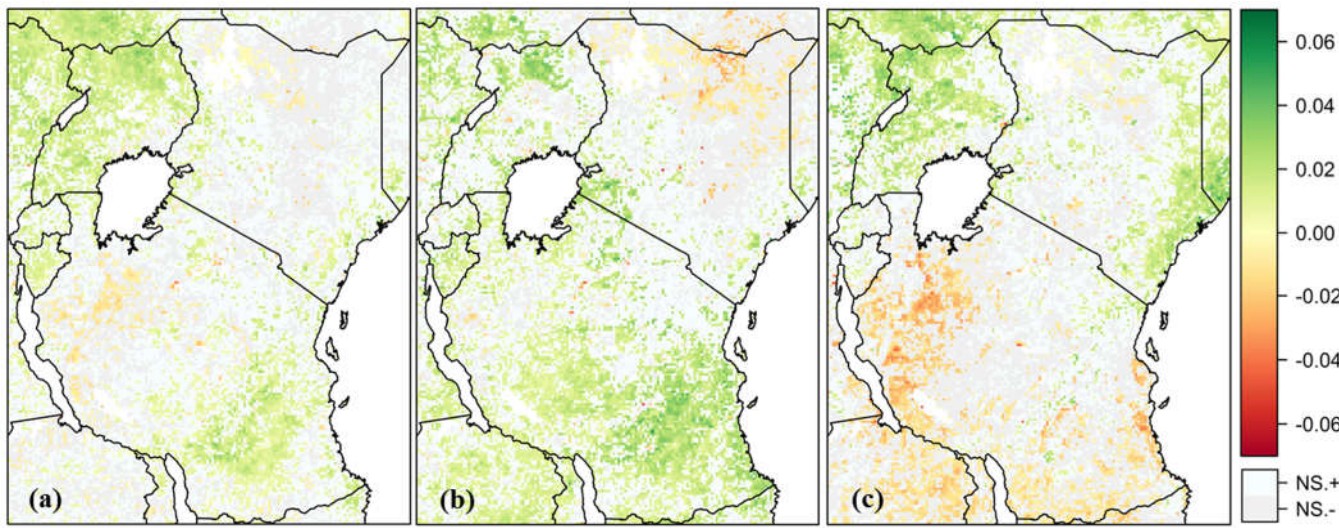

**Figure 2: Spatial patterns of long term (a) annual, (b) MAM, and (c) OND LAI trends. Significance of the trends is based on 95% confidence level. NS+ and NS- represents the non-significant positive and negative trends, respectively.**

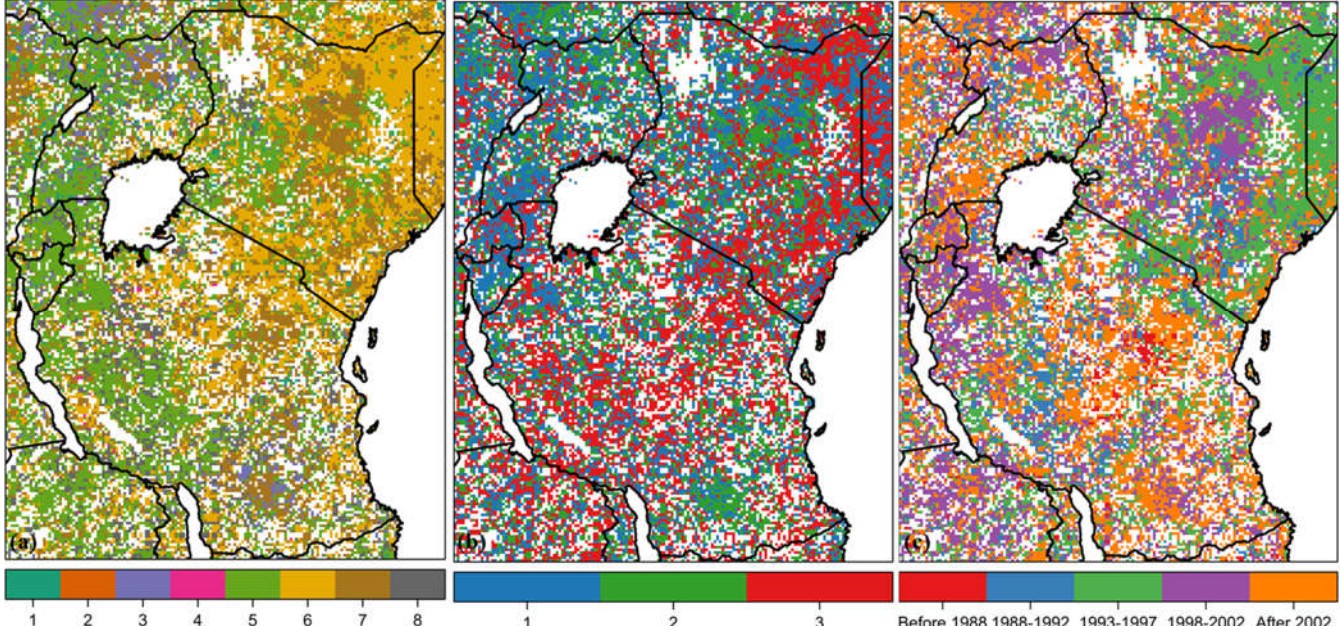

**Figure 3: (a) Type, (b) significance and (c) timing of trend shift in monthly LAI time series. Trends and breaks are considered as significant when P-value is below 0.05. Pixels with no significant (P < 0.05) change for all segments and/or no significant (P < 0.05) breakpoint are not shown. The trend shifts types in (a) are: (1) monotonic increase, (2) monotonic decrease, (3) monotonic increase (with positive break), (4) monotonic decrease (with negative break), (5) interruption: increase with negative break, (6) interruption: decrease with positive break, (7) reversal: increase to decrease, and (8) reversal: decrease to increase. The significance classes are: (1) both segments significant (or no break and significant), (2) only first segment significant, and (3) only 2nd segment significant.**





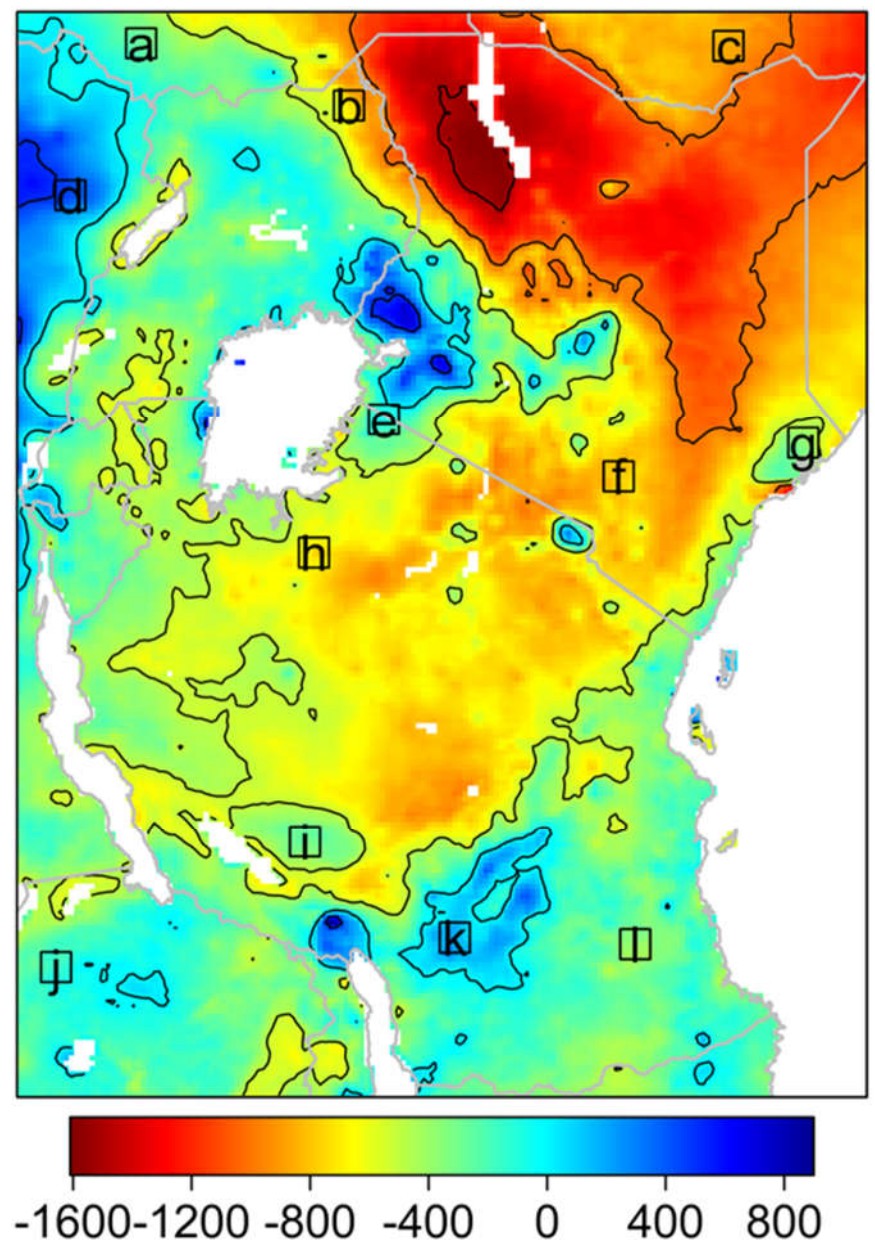

**Figure 4: Spatial distribution of average annual water balance during the period of 1982–2011. Regions circled by thick solid box are denoted as the typical water balance regions selected for case studies on LAI-SPEI correlation at various timescales.**



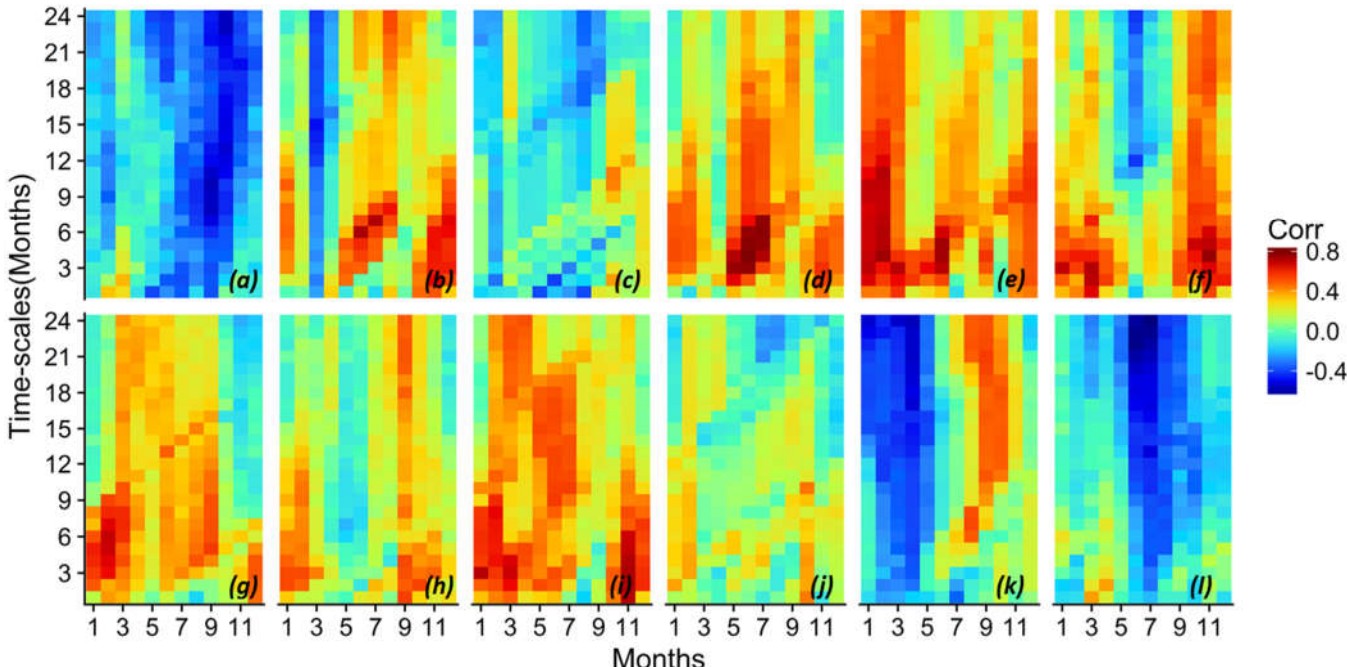

**Figure 5: Spearman correlation coefficient between LAI and SPEI at time scales from 1 to 24 months in the case study locations shown in Figure 4.**

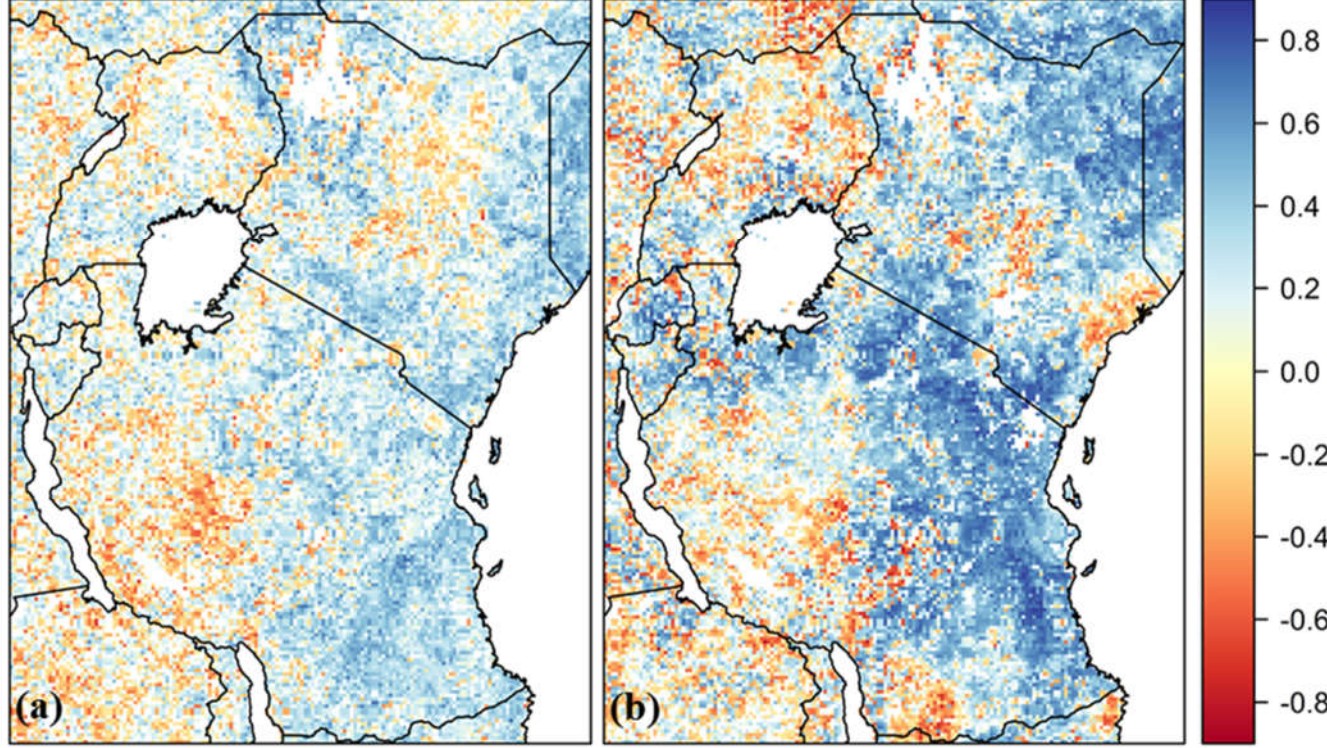

**Figure 6: Spatial overview of the Kendall τ coefficient for (a) LAI standard deviation time series derived over a twelve-year running window and (b) correlation between the LAI and SPEI twelve-year running window standard deviation time series. Only significant pixels are shown.**





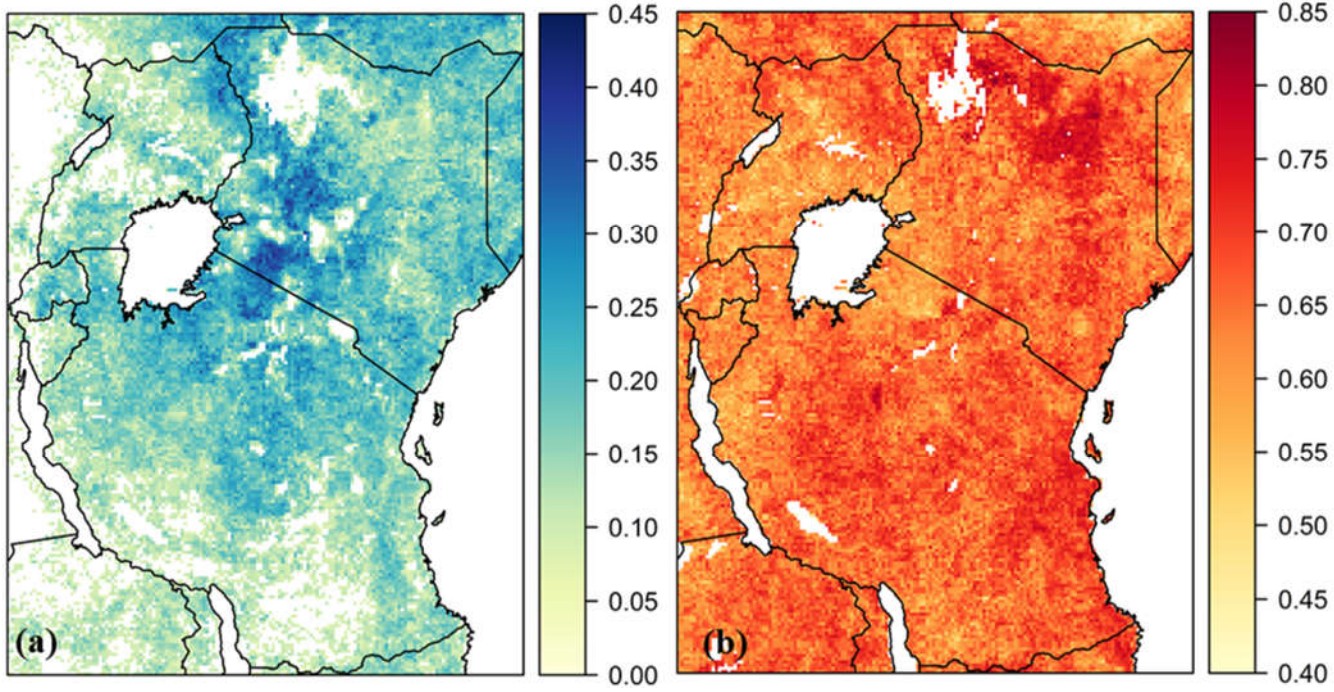

**Figure 7: Spatial patterns of vegetation (a) drought-resistance coefficient and (b) resilience coefficient. The pixels with insignificant coefficients are masked.**



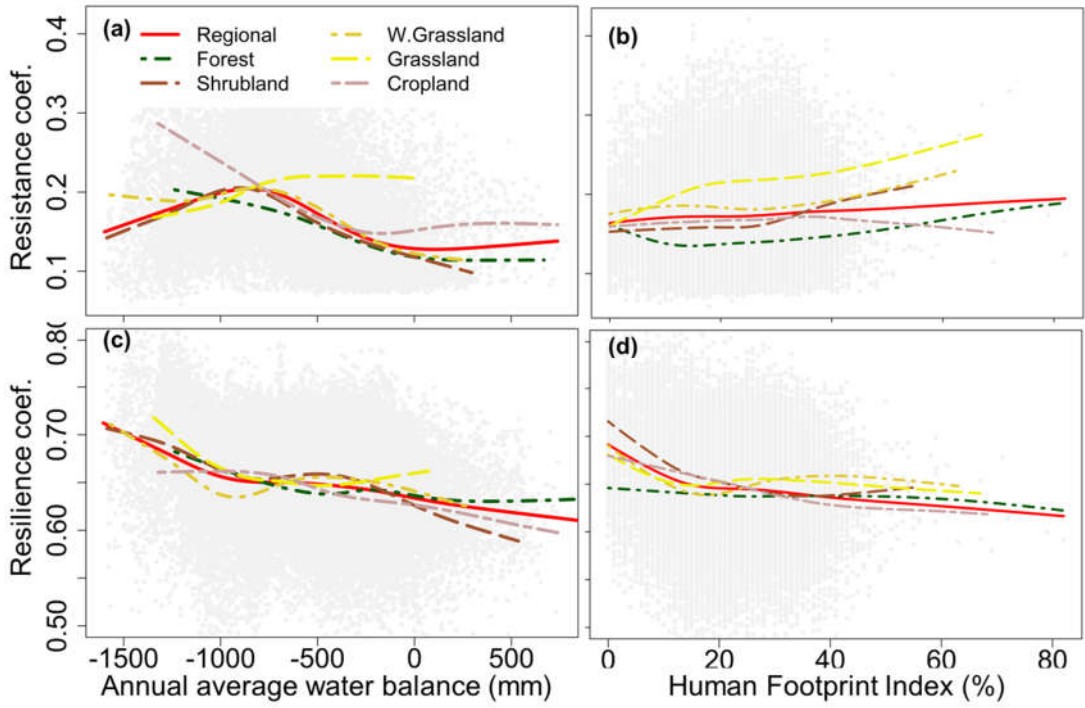

**Figure 8: The effect of mean annual water balance and human footprint index on the inter-annual vegetation response to SPEI anomalies. The local response in different land cover types is compared to the overall curve for East Africa (red line).**



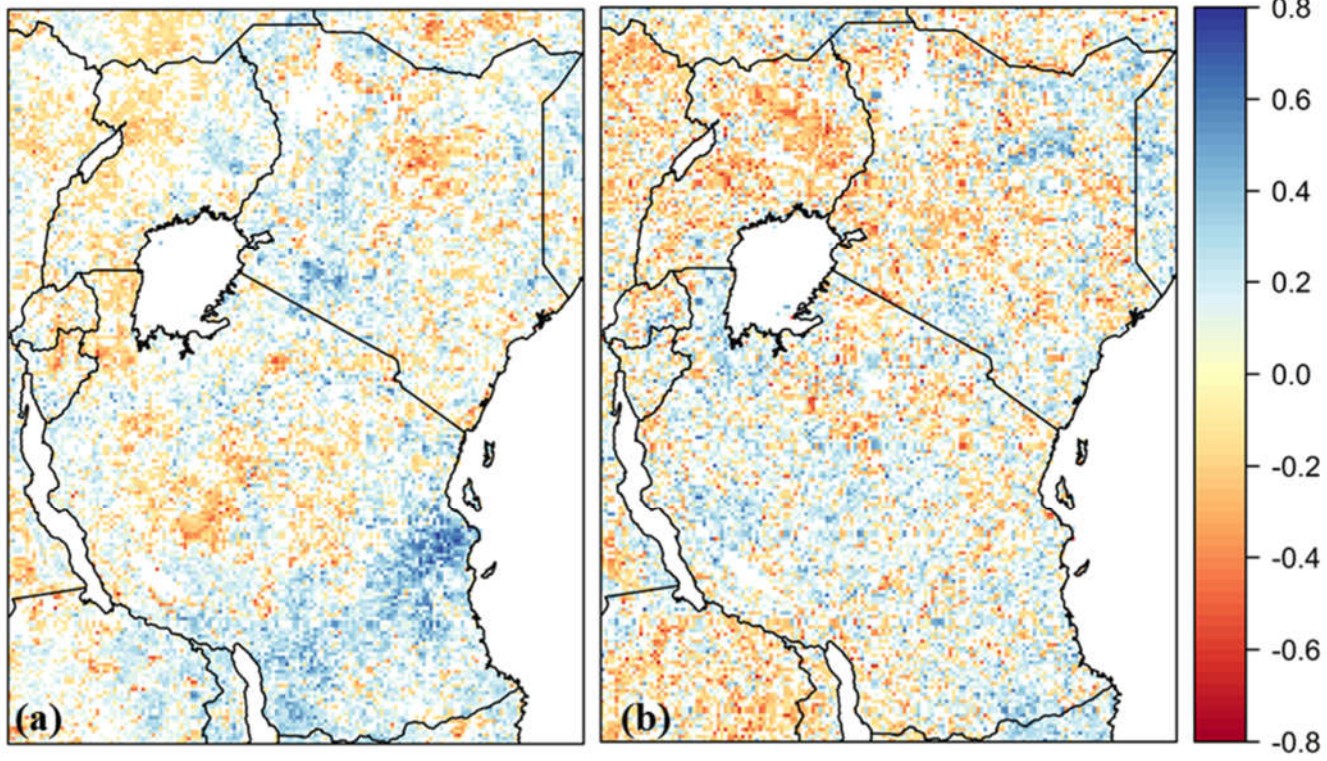

**Figure 9: Spatial overview of Kendall $\tau$ coefficient for (a) vegetation drought-resistance coefficient, and (b) vegetation resilience coefficient for the period 1982-2011. Only significant pixels are shown.**

