# Peer review of "Vegetation dynamics and responses to climate anomalies in East Africa"

_Earth System Dynamics, 2017_

## Referee Comment (RC1) · Anonymous Referee #1 · 21 Feb 2018

Overall, from a technical point of view, this work consists of solid data processing and statistical analysis methods for gridded time series. Applied methodologies are well-described and the choices for particular approaches and techniques are generally sufficiently justified, but it does not aim methodological advancement by comparative evaluation of methods, quantitative validation or development of novel techniques. However, from a thematic (environmental) angle, it lacks prior hypotheses on the environmental mechanisms that are underlying to the time series models being tested in the absence of hypotheses the results are presented in a very descriptive way and lack interpretation and synthesis. Only towards the end of the paper, some hints as to the environmental processes at work are suggested and are inductively constructed from sets of positive and negative regression coefficients, where these coefficients are

treated as data rather than parameters to evaluate and interpret models.

Therefore, the article in its current form does not fully qualify either as a methodological novelty, nor does it present conclusive thematic insights into the role of climate variability in the recent evolution of ecosystems and managed land use systems. I recommend it to be reworked towards either of these directions: A) a systematic evaluation of a new methodology to extract environmental metrics from spatiotemporal data with a thematic case study on East Africa or B) an in-depth thematic study on the variability of climate conditions in East Africa and the mechanistic effects on a different range of ecosystems, modulated by human management. I feel it has most potential for option A, but that would require a stronger emphasis on the novelty of your sequence of extraction methods, and a way to quantitatively evaluate how these methods perform compared to baseline methods or studies.

See attached report for specific comments to the manuscript sections.

Please also note the supplement to this comment:
https://www.earth-syst-dynam-discuss.net/esd-2017-123/esd-2017-123-RC1-supplement.pdf

**Supplement:**

Manuscript "Vegetation dynamics and responses to climate anomalies in East Africa"

1. Overall appreciation

This paper presents a spatiotemporal study of trends in and interactions between terrestrial vegetation in different East African ecosystems (represented by LAI) and climate variability (mainly represented by a form precipitation anomalies, SPEI). Quantitative metrics are extracted through (i) the estimation of temporal trends within time series of the individual variables, (ii) a lagged-response model the interaction between them, and (iii) a spatial analysis of the responses in terms of environmental variables. The results are interpreted in terms of earlier documented regional cases.

Overall, from a technical point of view, this work consists of solid data processing and statistical analysis methods for gridded time series. Applied methodologies are well-described and the choices for particular approaches and techniques are generally sufficiently justified, but it does not aim methodological advancement by comparative evaluation of methods, quantitative validation or development of novel techniques. However, from a thematic (environmental) angle, it lacks prior hypotheses on the environmental mechanisms that are underlying to the time series models being tested in the absence of hypotheses the results are presented in a very descriptive way and lack interpretation and synthesis. Only towards the end of the paper, some hints as to the environmental processes at work are suggested and are inductively constructed from sets of positive and negative regression coefficients, where these coefficients are treated as data rather than parameters to evaluate and interpret models.

Therefore, the article in its current form does not fully qualify either as a methodological novelty, nor does it present conclusive thematic insights into the role of climate variability in the recent evolution of ecosystems and managed land use systems. I recommend it to be reworked towards either of these directions: A) a systematic evaluation of a new methodology to extract environmental metrics from spatiotemporal data with a thematic case study on East Africa or B) an in-depth thematic study on the variability of climate conditions in East Africa and the mechanistic effects on a different range of ecosystems, modulated by human management. I feel it has most potential for option A, but that would require a stronger emphasis on the novelty of your sequence of extraction methods, and a way to quantitatively evaluate how these methods perform compared to baseline methods or studies.

2. Specific comments per section in the manuscript

Introduction

P1.L23-P2.L3: In this paragraph, the two-way interaction between terrestrial vegetation and climate dynamics is suggested by a series of statements that mention the one-way effects as well as the notion of interaction. On P2.L19-24 some of this information is repeated, which is redundant. It is also not concluded towards which effect will particularly be investigated,

although the remainder of the study uses one-way effect of climate on vegetation as a working hypothesis.

P2.L26-31: Some (numerically) detailed figures are given on land use conversions, but it is not clear why this is relevant, as the anthropogenic factor is only approximated as an environmental factor by the Human Footprint Index.

Data

P2.L11-14: Why are the time series smoothed? The 'superfluous values' argument is not clear. What is the rationale behind Maximum Value Composition? For NDVI, it is known that most atmospheric disturbance effects pose a negative bias on the NDVI, is this also the case for LAI?

P2.L25-P3.L3: What does the grouping of LC mean in terms of ecosystem stratification? For example, why are 'shrub-barren mosaic' and 'shrubs-grasses mosaic' in different groups, each with their tree-rich variant?

P3.L4-6: The description of the Human Footprint Index is insufficient: it is not clear what these data represent and what their role is in the analysis.

Methods

P4.L3-4: "The resultant per-pixel anomalies for each dataset were then averaged to obtain region-wide average time series". Are per-pixel anomalies z-scores or absolute deviations from the 30y-mean? In both cases, is it meaningful to average per-pixel anomalies? It seems to make more sense to differencing/z-scoring after averaging.

P7.L23: Why specifically a 12-year window?

P7.L30-31. Water balance should be described in the data section.

P8.L1-2: "Local effects are analysed per land cover type" What is the definition of 'local' here? The 6 grouped land cover strata still represent vaste areas.

Results

P8.L6-11: There is a serious issue with reporting annual change rates of 0.004 LAI units and 0.035°C. Although these may come out of the trend models as statistically different from zero, the effect sizes must also be interpreted the light of numerical precision and physical meaning. I do not believe LAI is measured with a 0.001 precision or temperature resampled to 8 km can be more precise than 0.1°. With proper rounding, the annual trend magnitudes are effectively zero!

P8.24-P9.21: The results are very descriptive, in terms of 'numbers' and 'proportions' of pixels that have either positive or negative values. I think if these observations are synthesized to effects playing in particular, meaningful ecoregions, they would be a lot more readable. Also,

the categories of break models (reverse, decreasing with positive break, etc.) lack a clear link to the environmental processes they represent. What is the difference between 'increase with negative break' and 'increase to decrease' in LAI for a savannah grassland ecosystem and what is the role of climate variability in year-to-year differences in greening of savannah?

P10.L1-21: Some of these descriptions illustrate the statement in 'Overall appreciation' that some hints to possible mechanisms are induced from statistical parameters, e.g., L19-21: "Stronger positive trends in vegetation show a similar pattern compared to the LAI-SPEI correlation, implying widespread influence of precipitation on vegetation trends in the region" The question remains what 'influence' exactly means here? This reads like statistics with hypothesis.

P11.25-27: "lower sensitivity to human influence below 35% of the human footprint index": it is absolutely not clear what this threshold on this index precisely represents, or what can be concluded from the observation that drought resistance of vegetation appears to be lower where this index is below 35%.

Discussion + Conclusions

I think these sections requires more in-depth synthesis. Most of the interpretations are formulated as 'phenomenom X varies spatially / shows contrasting spatial patterns / show spatial heterogeneity' which does not provide the reader with insights on what climate-coupled changes have or have not occurred in these ecosystems. The conclusions section is only a brief summary of introduction, data and methods and does not touch the results or their interpretation.

P12.L18-19: "This increase in LAI could be attributed to increasing land cover transition to croplands"? Transition from which land cover?

P13.32-33: "The temporal changes in the vegetation response metric imply technological and ecological effects". What does this mean? Please clarify.

Figures

Figure 3: I am aware that it is not easy to represent many spatiotemporal dimensions and metrics on graphs. For panels (a) and (b), the legends should contain the actual values, rather than a numbering that refers to the figure caption text.

Figure 5: The caption does not contain sufficient information to understand the graphs and what they represent.

3. Textual suggestions

P3.L26-28: I think 'blended' here means 'added'? In that case, I would suggest to uses 'added'

P4.L15-17: There is a text issue with the numbering and listing of the classes. Text here says 6 classes, the numbering in text jumps from (iii) to (vi) and the legend of Figure 3 you have 8 classes.

---

## Referee Comment (RC2) · Anonymous Referee #2 · 21 Feb 2018

Although not ground breaking, this paper presents a solid co-analysis of available long time series of LAI and SPEI with reference to the Global Human Footprint index in East-Africa.

What is disturbing though is that the abstract states: "Much-needed reference point for the disentanglement of historical climatic- and human-induced vegetation dynamics" while a p.14, L3 it is stated: "However, further analysis is required to disentangle human and climatic induced causes of these variations"

More thorough discussion of the disentanglement of vegetation dynamics into a human-induced and climatic-induced contributions is necessary.
* * *
[Figure]

2018.

---

## Author Comment (AC1) · 19 Mar 2018

**Author Response to Reviewer #1:**

We would like to thank the reviewer for providing detailed comments on our manuscript. Below, we have responded to each review comment (in blue font).

1. Overall appreciation

This paper presents a spatiotemporal study of trends in and interactions between terrestrial vegetation in different East African ecosystems (represented by LAI) and climate variability (mainly represented by some form precipitation anomalies, SPEI). Quantitative metrics are extracted through (i) the estimation of temporal trends within time series of the individual variables, (ii) a lagged-response model the interaction between them, and (iii) a spatial analysis of the responses in terms of environmental variables. The results are interpreted in terms of earlier documented regional cases.

Overall, from a technical point of view, this work consists of solid data processing and statistical analysis methods for gridded time series. Applied methodologies are well-described and the choices for particular approaches and techniques are generally sufficiently justified, but it does not aim methodological advancement by comparative evaluation of methods, quantitative validation or development of novel techniques. However, from a thematic (environmental) angle, it lacks prior hypotheses on the environmental mechanisms that are underlying to the time series models being tested in the absence of hypotheses the results are presented in a very descriptive way and lack interpretation and synthesis. Only towards the end of the paper, some hints as to the environmental processes at work are suggested and are inductively constructed from sets of positive and negative regression coefficients, where these coefficients are treated as data rather than parameters to evaluate and interpret models.

Therefore, the article in its current form does not fully qualify either as a methodological novelty, nor does it present conclusive thematic insights into the role of climate variability in the recent evolution of ecosystems and managed land use systems. I recommend it to be reworked towards either of these directions: A) a systematic evaluation of a new methodology to extract environmental metrics from spatio-temporal data with a thematic case study on East Africa or B) an in-depth thematic study on the variability of climate conditions in East Africa and the mechanistic effects on a different range of ecosystems, modulated by human management. I feel it has most potential for option A, but that would require a stronger emphasis on the novelty of your sequence of extraction methods, and a way to quantitatively evaluate how these methods perform compared to baseline methods or studies.

**Response**: Our work's main focus was on the vegetation trends and the extraction of metrics for vegetation resilience and resistance to short-term climate anomalies in East Africa. Nonetheless, we do agree with the reviewer on the need to: (1) provide a clearer statement of the objectives of the study in the Introduction section, and (2) emphasize in the Methods section on the novelty of the sequence of extraction methods that we adopted.

In the revised manuscript, in addition to clear statement of study objectives, the methodological framework was improved by: (i) providing a thorough inspection of regional ecosystems, (ii) assessment of the impacts of water memory effect on the vegetation resilience and (iii) using a non-linear regression approach to quantitatively evaluate the significance of the AR(1) model.

2. Specific comments per section in the manuscript

Introduction
P1.L23-P2.L3: In this paragraph, the two-way interaction between terrestrial vegetation and climate dynamics is suggested by a series of statements that mention the one-way effects as well as the notion of interaction.

**Response:** Our analysis only aimed to address the one-way interaction between vegetation and climate anomalies. In the revised manuscript, we modified the paragraph to explicitly state this.

On P2. L19-24 some of this information is repeated, which is redundant. It is also not concluded towards which effect will particularly be investigated, although the remainder of the study uses one-way effect of climate on vegetation as a working hypothesis.

**Response:** In the revised manuscript, we explicitly stated that our focus is only on the one-way interaction between vegetation and climate anomalies. We will also modify the paragraph to ensure that there is no repetition of information.

P2.L26-31: Some (numerically) detailed figures are given on land use conversions, but it is not clear why this is relevant, as the anthropogenic factor is only approximated as an environmental factor by the Human Footprint Index.

**Response:** The indication of the land cover changes was intended to put emphasis on the potential human-induced changes on the vegetation dynamics in the region. In the revised manuscript, we provided a clear link between these values and the human footprint index.

Data
P2.L11-14: Why are the time series smoothed? The 'superfluous values' argument is not clear. What is the rationale behind Maximum Value Composition? For NDVI, it is known that most atmospheric disturbance effects pose a negative bias on the NDVI, is this also the case for LAI?

**Response:** The LAI dataset used in this study was generated based on a neural network algorithm developed between the new improved third generation Global Inventory Modeling and Mapping Studies (GIMMS) Normalized Difference Vegetation Index (NDVI3g) and best-quality Terra Moderate Resolution Imaging Spectroradiometer (MODIS) LAI products for the overlapping period 2000–2009. Although the algorithm was trained on monthly data obtained by averaged biweekly values in every month, the 15-day NDVI data used in the LAI production was

not smoothed. The smoothing in this study was therefore aimed to reduce residual atmospheric noise propagated for the NDVI dataset. The Maximum Value Composite approach was appropriate due to the negative effects of atmospheric noise on the NDVI.

In the revised manuscript, we added the above explanation to clarify the need for data smoothing.

P2.L25-P3.L3: What does the grouping of LC mean in terms of ecosystem stratification? For example, why are 'shrub-barren mosaic' and 'shrubs-grasses mosaic' in different groups, each with their tree-rich variant?

**Response:** The land cover product used in the study is based on fuzzy logic approach exploiting synergies of multiple global land cover products. Affinity scores were used to link defined legend classes with the legend classes of the original products whereby the indicated land cover class covers more than 50% of the pixel; while in the case of mixed classes the indicated class combination is maximal relative to all other class possibilities. In our grouping of land cover classes, the simple legend defined in the reference article for the synergetic land cover data was used for purposes of making the land cover products comparable in terms of classification schemes.

In the revised manuscript, we included the above explanation in the Methods section.

P3.L4-6: The description of the Human Footprint Index is insufficient: it is not clear what these data represent and what their role is in the analysis.

**Response:** In the revised manuscript, we provided more details on the Human Footprint Index and its value for our analysis.

Methods
P4.L3-4: "The resultant per-pixel anomalies for each dataset were then averaged to obtain region-wide average time series". Are per-pixel anomalies z-scores or absolute deviations from the 30y-mean? In both cases, is it meaningful to average per-pixel anomalies? It seems to make more sense to differencing/z-scoring after averaging.

**Response:** Thank you for this suggestion. In the revised manuscript, we addressed this issue by calculating z-scores after averaging.

P7.L23: Why specifically a 12-year window?

**Response:** A 12-year window was used in the extraction of time-varying regression coefficients to ensure adequate size of time series for the regression analysis. A 12-year window implies 144 length of time series per variable which was considered adequate for ensuring statistical

significance of the coefficients. A similar window size was adopted by De Keersmaecker et al., (2017)

P7.L30-31. Water balance should be described in the data section.

**Response:** Thank you for this suggestion. In the revised manuscript, we provided more details on water balance.

P8.L1-2: "Local effects are analysed per land cover type" What is the definition of 'local' here? The 6 grouped land cover strata still represent vaste areas.

**Response:** The reviewer is correct in noting that the area covered by different land cover types is vast and that the use of the term "local effects" is not clear. In the revision, that sentence is changed to "The regional effect of these factors on vegetation response is modelled using data across the study area while affects in specific land cover types are analysed by considering areas consisting of common land cover types."

Results
P8.L6-11: There is a serious issue with reporting annual change rates of 0.004 LAI units and 0.035°C. Although these may come out of the trend models as statistically different from zero, the effect sizes must also be interpreted the light of numerical precision and physical meaning. I do not believe LAI is measured with a 0.001 precision or temperature resampled to 8 km can be more precise than 0.1°. With proper rounding, the annual trend magnitudes are effectively zero!

**Response:** In the revised manuscript, we also reported the trend values per decade to address this issue.

P8.24-P9.21: The results are very descriptive, in terms of 'numbers' and 'proportions' of pixels that have either positive or negative values. I think if these observations are synthesized to effects playing in particular, meaningful ecoregions, they would be a lot more readable. Also, the categories of break models (reverse, decreasing with positive break, etc.) lack a clear link to the environmental processes they represent. What is the difference between 'increase with negative break' and 'increase to decrease' in LAI for a savannah grassland ecosystem and what is the role of climate variability in year-to-year differences in greening of savannah?

**Response:** Thank you for highlighting this issue. In the revised manuscript, we modified the Results section as per the reviewer's suggestions.

P10.L1-21: Some of these descriptions illustrate the statement in 'Overall appreciation' that some hints to possible mechanisms are induced from statistical parameters, e.g., L19-21: "Stronger positive trends in vegetation show a similar pattern compared to the LAI-SPEI

correlation, implying widespread influence of precipitation on vegetation trends in the region"
The question remains what 'influence' exactly means here? This reads like statistics with
hypothesis.

**Response:** The term 'influence' is used in this context to mean correlation and therefore imply a
potential causation.

P11.25-27: "lower sensitivity to human influence below 35% of the human footprint index": it is
absolutely not clear what this threshold on this index precisely represents, or what can be
concluded from the observation that drought resistance of vegetation appears to be lower where
this index is below 35%.

**Response:** The human footprint index is used in this study as a proxy for human influence on the
vegetation patterns. The main idea of the footprint score is not to determine ecosystem hotspot
areas, but to highlight areas where ecosystems are expected to face disturbances, with either
potential and already observed impacts. The 35% reported in the sensitivity of vegetation
resilience to the index is only reported as a result of the patterns in the correlation between
resilience coefficient and the footprint index.

In the revised manuscript, a detailed explanation was provided to clarify the meaning of the
scores in the index.

Discussion + Conclusions

I think these sections requires more in-depth synthesis. Most of the interpretations are formulated
as 'phenomenom X varies spatially / shows contrasting spatial patterns / show spatial
heterogeneity' which does not provide the reader with insights on what climate-coupled changes
have or have not occurred in these ecosystems. The conclusions section is only a brief summary
of introduction, data and methods and does not touch the results or their interpretation.

**Response:** Thank you for these suggestions.  In the revised manuscript, the sections were
modified to provide better insights into the climate-coupled changes of the vegetation response.

P12.L18-19: "This increase in LAI could be attributed to increasing land cover transition to
croplands"? Transition from which land cover?

**Response:** In the revised manuscript, we modified it to "This increase in LAI could be attributed
to increasing land cover transition to croplands from grassland"

P13.32-33: "The temporal changes in the vegetation response metric imply technical and
ecological effects". What does this mean? Please clarify.

**Response:** Although a bit of explanation is provided in the sentences following this statement, in
the revised manuscript, we provided a better explanation.

The technical implications of the non-stationarity in the vegetation stability metrics points to the need to consider break-points in vegetation time series or the explicit inclusion of temporal changes in the stability coefficients as well as the effects of the time frame and time series length considered in the analysis. On the other hand, ecological implications points to the need for consideration of changes in vegetation structure and functional types through monitoring of vegetation changes particularly through land use changes and land degradation.

Figures

Figure 3: I am aware that it is not easy to represent many spatio-temporal dimensions and metrics on graphs. For panels (a) and (b), the legends should contain the actual values, rather than a numbering that refers to the figure caption text.

**Response:** In the revised manuscript, we provided a full description of the classes and, if need be, the figures will be provided as supplementary figures.

Figure 5: The caption does not contain sufficient information to understand the graphs and what they represent.

**Response:** In the revised manuscript, we provided more detailed information in the caption.

2. Textual suggestions
P3.L26-28: I think 'blended' here means 'added'? In that case, I would suggest to uses 'added'

**Response:** Thank you for this suggestion. In the revised manuscript, we modified this text to 'added'.

P4.L15-17: There is a text issue with the numbering and listing of the classes. Text here says 6 classes, the numbering in text jumps from (iii) to (vi) and the legend of Figure 3 you have 8 classes.

**Response:** Thank you for highlighting this inconsistency. In the text, classes 3 and 4 (interrupted monotonic trends) were considered monotonic changes. In the revised manuscript, this issue was addressed.

**References**
De Keersmaecker, W., Lhermitte, S., Hill, M. J., Tits, L., Coppin, P., & Somers, B. (2017). Assessment of regional vegetation response to climate anomalies: A case study for australia using GIMMS NDVI time series between 1982 and 2006. *Remote Sensing*, *9*(1), 1–17. https://doi.org/10.3390/rs9010034

---

## Author Comment (AC2) · 19 Mar 2018

**Author Response to Reviewer #2:**

We would like to thank the reviewer for providing detailed comments on our manuscript. Below, we have responded to each review comment (in blue font).

Although not ground breaking, this paper presents a solid co-analysis of available long time series of LAI and SPEI with reference to the Global Human Footprint index in East-Africa. What is disturbing though is that the abstract states: "Much-needed reference point for the disentanglement of historical climatic- and human-induced vegetation dynamics" while a p.14, L3 it is stated: "However, further analysis is required to disentangle human and climatic induced causes of these variations" More thorough discussion of the disentanglement of vegetation dynamics into a human-induced and climatic-induced contributions is necessary.

Response: The main focus of this study was the analysis of trends in vegetation based on LAI time series and the response to climate anomalies. Although our analysis did not address the disentanglement of vegetation dynamics into human-induced and climatic-induced contributions, this was identified as an area for future study. In the revision, a clear discussion was provided on the influence of the human footprint on the vegetation responses to climate anomalies.